# Effect of Running Velocity Variation on the Aerobic Cost of Running

**DOI:** 10.3390/ijerph18042025

**Published:** 2021-02-19

**Authors:** Madeline Ranum, Carl Foster, Clayton Camic, Glenn Wright, Flavia Guidotti, Jos J de Koning, Christopher Dodge, John P. Porcari

**Affiliations:** 1Department of Exercise and Sports Science, University of Wisconsin, La Crosse, WI 54601, USA; madey.ranum@gmail.com (M.R.); gwright@uwlax.edu (G.W.); cdodge@uwlax.edu (C.D.); jporcari@uwlax.edu (J.P.P.); 2Kinesiology and Physical Education, Northern Illinois University, De Kalb, IL 60115, USA; cccamic1@uni.edu; 3Institute of Motor Sceinces, University of Rome-Foro Italico, 00135 Rome, Italy; guidotti.flavia@gmail.com; 4Faculty FBW, Vrije Universiteit, 1081 BT Amsterdam, The Netherlands; j.dekoning@fbw.vu.nl

**Keywords:** running, cost of running, running performance

## Abstract

The aerobic cost of running (CR), an important determinant of running performance, is usually measured during constant speed running. However, constant speed does not adequately reflect the nature of human locomotion, particularly competitive races, which include stochastic variations in pace. Studies in non-athletic individuals suggest that stochastic variations in running velocity produce little change in CR. This study was designed to evaluate whether variations in running speed influence CR in trained runners. Twenty competitive runners (12 m, VO_2max_ = 73 ± 7 mL/kg; 8f, VO_2max_ = 57 ± 6 mL/kg) ran four 6-minute bouts at an average speed calculated to require ~90% ventilatory threshold (VT) (measured using both v-slope and ventilatory equivalent). Each interval was run with minute-to-minute pace variation around average speed. CR was measured over the last 2 min. The coefficient of variation (CV) of running speed was calculated to quantify pace variations: ±0.0 m∙s^−1^ (CV = 0%), ±0.04 m∙s^−1^ (CV = 1.4%), ±0.13 m∙s^−1^(CV = 4.2%), and ±0.22 m∙s^−1^(CV = 7%). No differences in CR, HR, or blood lactate (BLa) were found amongst the variations in running pace. Rating of perceived exertion (RPE) was significantly higher only in the 7% CV condition. The results support earlier studies with short term (3s) pace variations, that pace variation within the limits often seen in competitive races did not affect CR when measured at running speeds below VT.

## 1. Introduction

As defining features of human performance, world record (WR) performances have been of interest to physiologists. A number of physiological determinants have emerged as indicators of performance. These determinants include maximal aerobic capacity, fractional utilization of aerobic capacity, lactate threshold, and the aerobic cost of running (CR) [1]. The VO_2_ or velocity at the ventilatory threshold (VT) has also been shown to predict running performance both cross-sectionally and longitudinally [2]. Pacing strategy has also emerged as a potential determinant of performance, as contemporary records have been achieved with a relatively more even [3] or unchanged pacing strategy [4], although important head-to-head competitive events are typically contested with highly stochastic pacing patterns [5]. Given that VO_2_max has largely been unchanged in elite runners over the last half-century, and evidence is lacking on systematic changes in anaerobic capacity, only VT, fractional utilization and CR provide reasonable candidates for the continuing improvement in running records. Fractional utilization is of established importance within competitive runners [6] and is particularly high in the most elite runners [7,8]. The CR is thought to vary meaningfully amongst runners and is particularly low in elite runners [8,9]. Although inadequately documented, the traditional view is that changes in velocity are “expensive” in terms of the CR [10]. Studies of intermittent work suggest that the anaerobic capacity, represented by W’, can be depleted more rapidly than it can be repaid [11], which might provide further rationale for the emergence of a relatively more constant velocity profile in record setting races [3]. Recent evidence suggests that the D’, a surrogate of W’ within the critical power/critical speed paradigm, is nearly perfectly expended at the end of one-mile world record races, which argues that widely uneven pacing patterns may be disadvantageous [12]. However, Minetti et al. [13] have shown that even relatively large short-term variations in running velocity do not have a large impact on the net cost of running. However, the magnitude of the velocity variations studied by Minetti et al. [13] was very large (10–40%) compared to the lap-to-lap velocity variation in competitive runners (2–7%) [3,4,5]. Accordingly, this study was designed to determine the effect of pace variation (around a constant net running velocity) on CR and physiological responses in competitive runners. Recognizing the substantial problems of estimating the aerobic cost at velocities above those that allow a steady state of VO_2_ (particularly in elite runners) [14], we have taken the approach that if velocity variations are important as drivers of energy cost generally, they will be evident at speeds allowing VO_2_ to represent energy cost in sub-elite runners. 

## 2. Materials and Methods

### 2.1. Participants

The subjects in this study were 20 competitive runners (men n = 12, women n = 8) recruited from a NCAA Division III track team. All runners had an extensive running background and were fit and healthy at the time of testing (predominantly Fall after cross-country season or Winter during indoor track season). Approval was obtained from the university human subjects committee (protocol 45CFR46, 46-110, 2013) and each subject provided written informed consent prior to participation. Descriptive characteristics of the subjects are provided in Table 1. 

### 2.2. Experimental Design

Each subject performed two trials. Each trial was performed after a normal regeneration day (40–60 min of easy running) within the coach’s periodization scheme. Trial 1 was an incremental treadmill test to determine VO_2_max and ventilatory threshold (VT). The treadmill grade was constant at 1% to account for the lack of wind resistance in the laboratory [15]. The speed was incremented from walking (1.33 m·s^−1^) to maximal running in increments of 0.21 m·s^−1^ during each minute. At least 48 h later, each subject completed Trial 2 consisting of four, 6-minute running intervals with 2 min of walking at 1.33 m·s^−1^ between intervals. The average velocity of each run was set to require ~90% VT, to ensure that VO_2_ was likely to be in steady state. One interval was completed at a constant speed (control) while the other three intervals were run with the pace altered above and below the average speed each minute by 0.044 m·s^−1^, 0.134 m·s^−1^ or 0.223 m·s^−1^. For example, if the subject’s average speed was determined as 3.35 m·s^−1^ at ~90% VT, the constant-paced interval was run at 3.35 m·s^−1^ for 6 min. The varied pace intervals were run at 3.31/3.39, 3.22/3.48, and 3.13/3.57 m·s^−1^, respectively. Each of the varied pace intervals were 6 min in duration with velocity change occurring every minute. The order of sets was counterbalanced to account for thermal stress and fatigue. Within each set, the slower segment was run first. The VO_2_ during the last 2 min of each 6-minute set was accepted as CR.

### 2.3. Procedures

For all tests, VO_2_ was measured using open circuit spirometry with a metabolic cart (AEI Technologies, Bastrop, TX, USA). Calibration procedures were standard, a 3-L syringe to calibrate the pneumotach and a reference gas (~16% O_2_ and ~5% CO_2_) and room air to calibrate the gas analyzers. Heart rate was monitored using radio-telemetry (Polar Electro-Oy, Portsmith, NY, USA). Both VO_2_ and HR were recorded as the average over the last 2 min of each set. Blood lactate was measured in fingertip blood during walking following each 6-minute interval using dry chemistry (Lactate Pro, Cosmed, Concord, CA, USA). Ratings of perceived exertion (RPEs) were made using the Category-Ratio (0–10) Borg Scale [16] in the last 15 s of each 6-minute interval. The CR was expressed as the gross VO_2_ required to run 1 km (mL·kg^−1^·km^−1^). CR of each varied pace run was compared with the CR of the constant pace run. VT was estimated using both the v-slope and ventilatory equivalent method [17].

### 2.4. Data Analysis

The difference in CR, and other outcome measures, expressed in relation to the coefficient of variation (CV) of running speed within each run was evaluated using repeated measures ANOVA. Post-hoc tests were done using the Tukey Test. A *p*-value of < 0.05 was considered statistically significant. The Stata statistical software, version 141 (StatCorp, College Station, TX, USA) was used for statistical analysis. A preliminary analysis was run with gender as a co-variate. It was not significant relative to changes in CR by the CV of running speed. Accordingly, the primary analysis was based on the pooled results of male and female subjects. 

## 3. Results 

### Physiological Responses

The physiological responses to the varied and constant pace intervals are presented graphically in Figure 1. The pace variations were expressed in relation to the CV of running velocity to allow for individual differences in running speed and to show the extent of variability around each runner’s average speed at ~90%VT. The average CV for each pace variation was 0% for the constant speed, 1.4% for 0.044 m·s^−1^, 4.2% for 0.134, and 7.0% for 0.223 m·s^−1^. 

There were no significant differences between the varied and constant pace intervals for CR (229.7 ± 17.4, 229.7 ± 17.1, 228.0 ± 14.7 and 229.1 ± 16.7 mL·kg^−1^·km^−1^), HR (162 ± 3, 162 ± 4, 163 ± 5 and 164 ± 5 bpm) or blood lactate (2.0 ± 0.5, 1.9 ± 0.5, 1.9 ± 0.5 and 1.9 ± 0.4 mmol*L^−1^), for the control, 1.4, 4.2 and 7.0% CV of velocity, respectively. A significant difference (*p* = 0.0246) was found in RPE (3.0 ± 0.3, 3.0 ± 0.3, 3.0 ± 0.2 and 3.2 ± 0.2) with a pace variation at 7.0% CV.

## 4. Discussion

### 4.1. Overall Discussion

The primary purpose of this study was to investigate how variations in pace around a constant average speed affected CR. The study reflected the nature of competitive races in which runners typically change their speed for tactical purposes. [5] or due to spontaneous changes in pacing [2,3]. In order to do this, subjects completed 6-minute intervals in which the pace was changed every minute around an average speed that required ~90% VT (e.g., 10-km-marathon pace). The changes in speed (0.044, 0.134 and 0.223 m·s^−1^) coincided, in this sample of sub-elite runners, with a CV of 1.4 ± 0.1, 4.2 ± 0.4, and 7.0 ± 0.7 % of average velocity, respectively. A run with no changes in velocity (0% CV) served as a control. This magnitude of CV was of the same magnitude as often reported in races when runners are improving their best performances [2,3]. It was found that within this range of CV, there was no significant change in CR compared with constant pace running. These results agree with the earlier findings of Minnetti et al. [13], who studied less elite runners during very short-term (3 s) variations in velocity, with an effective CV ranging from 0 to 40%. In addition, the imposed pace variations had no significant effect on HR or blood lactate. RPE was significantly higher only during the 7% CV condition. The results do not support an increase in CR with an increase in the CV of velocity, at least within the constraints of a clearly submaximal running velocity (~90%VT). 

While the ability of this group of runners was far below the ability of elite runners, who might be capable of WR, and the relative speed chosen (<VT) was below competition velocity in this population, the results suggest that variations in running velocity, per se, do not inherently change the CR. Future studies, perhaps using more accomplished runners, or technology not dependent on the limits of pulmonary gas exchange might amplify our results.

The absolute value for CR (~225 mL kg^−1^ km^−1^) was comparatively high compared to that observed in more elite runners [7,8]. Given the relatively good values for VO_2_max within this sample of competitive sub-elite, runners (best performance was 85–90% of WR velocity) it seems reasonable to suggest that the CR was a large determinant of the performance level of these subjects. Other data from our laboratory have noted a normal range of variation in CR, suggesting the internal validity of the present data [18].

There was a significant increase in the RPE during the set with the largest CV (7%), suggesting that large variations in pace might have a deleterious effect. However, Veneman et al. [19] recently demonstrated a non-significantly (0.4%) faster time in a 10 km cycling time trial when 10% positive and negative variations in power output (PO) were inserted in alternating kilometers, compared to a self-paced trial. The RPE during the trial with imposed variations in pace had a reduced average RPE (6.1 ± 1.6) versus the self-paced (6.8 ± 1.2) trial, presumably because the alternating of motor units allowed for a lower perceptual cost. However, during cycling it is possible for velocity to remain high despite reductions in PO because of the effect of momentum, which is relatively less important during running. The Venenman et al. [19] trial was conducted at near competitive intensities for an event of 15–20 min duration, compared to the current running trial, suggesting that the response of variations in effort at higher intensity may be different than the lack of differences in CR observed in the present trial. However, at intensities greater than the VT, measurement of VO_2_ becomes less valid as a unitary measure of energy cost.

Minetti et al. [13] have argued that successive storage/release of energy during acceleration related to tendon function is unlikely to preserve the CR during pace variations. Certainly, in the circumstances of this study, with a much less acute variation in running pace, it seems very unlikely that energy storage in the tendons might contribute to the result. More reasonably, the relative slowness of VO_2_ kinetics, which are known to produce near steady state values for VO_2_ during interval training, seem a likely explanation for the observed lack of variation in CR.

### 4.2. Practical Applications

CR is a potentially important determinant of success in the performance of competitive runners. Given the tendency for recent WR, at least in shorter events, to be run with smaller values for CV of velocity [3], it would seem to be reasonable to suspect that within-event variations in velocity might increase CR. However, changes in the CV of running velocity are less evident in longer events where athletes are “running for time” [4], and where VO_2_ might more reasonably represent the CR than during events where the CR probably exceeds VO_2_max. However, in highly competitive events, there are often large stochastic variations in momentary running velocity [5]. Although reasonably likely to affect the anaerobic capacity of the competitors, we cannot speculate on how this might influence the CR. This study showed that small variations in pace, representing a CV of running speed from 0 to 7%, which is within the range observed in competitive races [3,4,5,8], do not significantly affect CR, HR or blood lactate, at least at velocities below the VT.

## 5. Conclusions

Supporting earlier findings in non-competitive runners [13], variations in the CV of running velocity, up to 7% do not seem to systematically affect the CR in well-trained runners. Accordingly, the relatively smaller CV observed in some contemporary WR races [3] cannot reasonably be attributed to optimization of the CR, although specific confirmation in better runners, at higher relative speeds awaits further study. 

## Figures and Tables

**Figure 1 ijerph-18-02025-f001:**
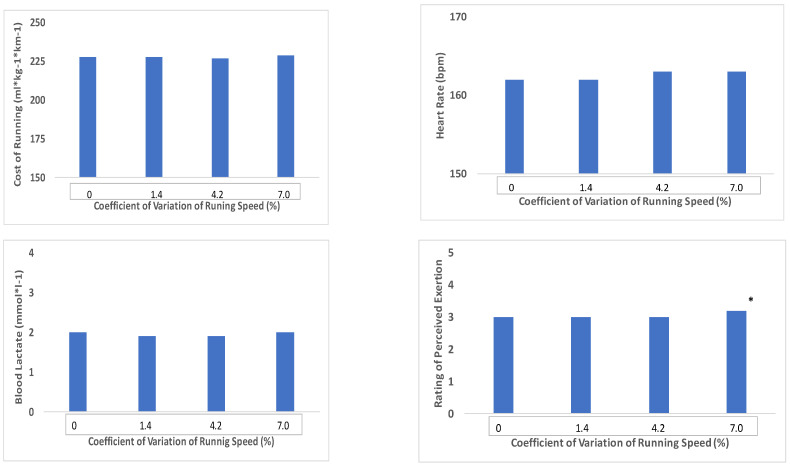
Cost of running expressed as mL·kg^−1^·km^−1^ (upper left), heart rate (upper right), blood lactate concentration (lower left), and rating of perceived exertion (RPE) (lower right) in relation to the coefficient of variation (CV) of running velocity, * equivalent to ×.

**Table 1 ijerph-18-02025-t001:** Characteristics (mean + sd) of the subjects.

Subjects	Men (n = 12)	Women (n = 8)	All (n = 20)
Age (years)	20.8 ± 1.9	21.6 ± 1.3	21.2 ± 1.7
Height (cm)	178.8 ± 5.6	168.1 ± 6.2	174.5 ± 7.8
Weight (kg)	69.0 ± 7.1	55.6 ± 6.3	63.7 ± 9.4
VO_2_max (mL/kg/min)	72.8 ± 6.8	57.1 ± 6.0	66.5 ± 10.1
VO_2_ @ VT (mL·kg^−1^·min^−1^)	53.5 ± 6.4	45.8 ± 4.1	50.4 ± 6.7
vVO_2_max (m·min^−1^)	326.1 ± 27.0	268.0 ± 17.5	302.8 ± 37.3
Velocity @ 90% VT (m·min^−1^)	223.3 ± 23.1	196.3 ± 10.2	212.5 ± 23.1
HRmax (bpm)	187.7 ± 7.3	185.3 ± 10.5	186.7 ± 8.4
Volume of Training (km·week^−1^)	84 ± 31	49 ± 24	69 ± 33
Average Pace (m·min^−1^)	225.2 ± 13.7	202.4 ± 12.7	216.1 ± 17.3
Interval Sessions·/week^−1^	0.8 ± 0.7	0.8 ± 0.9	0.8 ± 0.8

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
