# Peer review of "Effect of Running Velocity Variation on the Aerobic Cost of Running"

_ijerph, 2021, doi:10.3390/ijerph18042025_

Round 1
Reviewer 1 Report
This study compares oxygen cost of running at a constant pace (at 90% of ventilatory threshold) with variable pacing at three different magnitudes of variability around that same pace. The results suggest that the relatively minor fluctuations in pace do not significantly alter the cost of running. The authors have taken a sound methodological approach to comparing these 4 different conditions, including a 2 minute walk in between conditions, randomized order of the conditions, and averaging of the last 2 minutes of HR and VO2 data.
There are a few areas in which this paper needs to be improved in order to be acceptable for publication. The introduction and much of the discussion focuses on the importance of running economy for world record and other elite-level performance. While this is true, the present study examines cost of running at 90% of VT, a velocity that would appear to be a relatively easy run pace for the participants in this study. Thus, any application toward racing performance remains purely speculative. The idea of surges in a race vs. even pacing is a whole different topic, as those mid-race surges at an elite level will be at much higher speeds, and will likely include substantial anaerobic work. We appreciate that it is more challenging to calculate the cost of running at an above the anaerobic threshold. However, the present study simply doesn’t examine paces that are meaningful to the discussion of world record and elite level performance at 10k to the marathon, as even the marathon is run at faster speeds than the 90% of VT tested here. See: Tanaka and Matsuura (1984), Marathon performance, anaerobic threshold, and onset of blood lactate accumulation.
We suggest re-writing the introduction and discussion to reflect the lower speeds that this study examined. Relevant applications could be pacing in easy to moderate paced aerobic training runs, or possibly ultra-endurance performance if the authors wish to maintain a focus on elite performance. Other topics that should be included in the intro and discussion include previous research on VT and VO2 kinetics.
The following are additional areas for improvement: 1) Presenting the data more clearly, including means and standard deviations. It would also be useful to include RER data, or alternatively to calculate energy cost of running (see Fletcher, Esau, & MacIntosh, 2009. Economy of running: beyond the measurement of oxygen uptake). 2) Clarifying and providing more details on some of the methods, including determination of VT, perhaps a figure to show the testing protocol, and more details on the statistical analysis. 3) More robust citation throughout. 4) Correct minor grammatical errors and inconsistencies in units (ml/km/min some places, ml.km-1.min-1 others, for example).
See below for further line-by-line comments.
Abstract
Line 13 – remove “in” prior to “which”
Line 14-15 – make clear that variations need to be at a an exercise intensity below VT
Line 16 – remove or replace “in the range of 10km-marathon pace” as the speeds tested equate to roughly 7:15 min/mile for the men, which is an easy run pace for a collegiate distance runner
Line 19 – the use of the term “coefficient of variation” led me to believe that the variations in pace would involve a more complex fluctuation scheme. It would be more clear to simply say that paces were set to alternate each minute above and below 90% of VT by 1.4, 4.2, and 7.0%.
Include how VT was determined, and why this method was used.
Introduction
As noted above, the framing of the paper around world record performances is not appropriate given the methodology employed in this study. The introduction could use a paragraph on VT, as the use and determination of VT is a complicated topic in and of itself (VT1 vs. VT2, and the various methods of determining these thresholds). See the following for starters…
Ekkekakis, P., Lind, E., Hall, E. E., & Petruzzello, S. J. (2008). Do regression-based computer algorithms for determining the ventilatory threshold agree? Journal of sports sciences, 26(9), 967-976.
Faude, O., Kindermann, W., & Meyer, T. (2009). Lactate threshold concepts. Sports Medicine, 39(6), 469-490.
Methods:
Table 1: Are volumes of training correct? 30 km/week (or even 52 km/week for the men) seems very low for collegiate distance runners. If these are correct, it may be noteworthy and provide some explanation for the relatively poor running economy measured in this group.
Lines 68-69: Include a citation for the VO2max testing protocol, and perhaps a justification for using only treadmill speed instead of incline for the protocol.
This would also be a good place to include how VT was determined, along with a citation for the method of VT determination that was used.
Line 80-82: Clarify or consider adding a figure to clearly show the design.
Lines 86: State the specific model of metabolic system.
Line 92: Citation for Borg scale should be included.
Lines 97-99: Include info about software or code used for analysis. Also what was considered statistically significant?
Figure 1: Make x-axis titles consistent, consider adding dots for individual data to show variation. HR missing units in y-axis label. Y-axes could be re-scaled to better show variation between trials. Add symbol denoting significant difference in RPE figure.
Lines 113-117: move to discussion.
Line 113 – remove “comparatively”
We are also very curious to see these high CR values. Do you have other data from your lab to place this in context with your system, or other studies using the same system?
Discussion
Move discussion away from world record/elite performance, as discussed above.
Lines 120-129: too much redundant information summarizing the methods. 1-2 sentences would be sufficient before getting to the main findings (currently around line 129).
Line 155: VO2 kinetics is mentioned for the first time – this concept should be included in the introduction and expanded upon here.
Lines 170-171: “velocities consistent with steady state conditions” doesn’t make a lot of sense here, given that the whole ideas is that pace is varying. It seems more accurate to say “speeds consistent with moderate intensity exercise” or “speeds below the ventilatory threshold.”
Conclusions:
Again, this study does not provide any insight into pacing strategies in WR races due to the much lower intensity of exercise in this protocol.
Author Response
Comments and Suggestions for Authors
This study compares oxygen cost of running at a constant pace (at 90% of ventilatory threshold) with variable pacing at three different magnitudes of variability around that same pace. The results suggest that the relatively minor fluctuations in pace do not significantly alter the cost of running. The authors have taken a sound methodological approach to comparing these 4 different conditions, including a 2 minute walk in between conditions, randomized order of the conditions, and averaging of the last 2 minutes of HR and VO2 data.
There are a few areas in which this paper needs to be improved in order to be acceptable for publication. The introduction and much of the discussion focuses on the importance of running economy for world record and other elite-level performance. While this is true, the present study examines cost of running at 90% of VT, a velocity that would appear to be a relatively easy run pace for the participants in this study. Thus, any application toward racing performance remains purely speculative. The idea of surges in a race vs. even pacing is a whole different topic, as those mid-race surges at an elite level will be at much higher speeds, and will likely include substantial anaerobic work. We appreciate that it is more challenging to calculate the cost of running at an above the anaerobic threshold. However, the present study simply doesn’t examine paces that are meaningful to the discussion of world record and elite level performance at 10k to the marathon, as even the marathon is run at faster speeds than the 90% of VT tested here. See: Tanaka and Matsuura (1984), Marathon performance, anaerobic threshold, and onset of blood lactate accumulation.
REWRITTEN AS SUGGESTED
We suggest re-writing the introduction and discussion to reflect the lower speeds that this study examined. Relevant applications could be pacing in easy to moderate paced aerobic training runs, or possibly ultra-endurance performance if the authors wish to maintain a focus on elite performance. Other topics that should be included in the intro and discussion include previous research on VT and VO2 kinetics.
REWRITTEN AS SUGGESTED. WHILE THE ULTIMATE QUESTION DOES RELATED TO RECORD PERFORMANCES, WE WOULD POINT OUT THAT CHANGES IN COMPETITIVE PERFORMANCE, WHETHER IN ELITE RUNNERS OR RECREATIONAL RUNNER ARE OF THE SAME CHARACTER 4, AND THAT THE ISSUES INVOLVED IN ESTIMATING THE ENERGY COST OF RUNNING IN CONDITIONS WHERE A STEADY STATE OF OXYGEN UPTAKE ARE NOT AVAILABLE ARE SUBSTANTIAL. ACCORDINGLY, WE HAVE TAKEN THE VIEWPOINT THAT THE ADVANTAGES TO BE GAINED FROM MEASURING ENERGY COST AT VELOCITIES THAT ALLOW A STEAD STATE OF VO2 OUTWEIGH THE LOSS OF VELOCITY SPECIFICITY BY USING OTHER METHODS TO MEASURE ENERGY COST.14 THUS, WE HAVE TAKEN THE ATTITUDE THAT THE BEST PRACTICAL APPROACH TO ADDRESSING THE QUESTION OF WHETHER ELITE RUNNERS MIGHT HAVE DIFFERENCES IN RUNNING ECONOMY IN RELATION TO THE PACING PATTERN IS TO EVALUATE SUB ELITE RUNNERS AT VELOCITIES THAT ALLOW A STEADY STATE OF OXYGEN UPTAKE, ON THE GROUNDS THAT IF ACCELERATING AND DECELERATING ARE GENERALLY IMPORTANT, THAT THEY MIGHT BE SPECIFICALLY IMPORTANT AT COMPETITIVE VELOCITIES IN HIGH LEVEL RUNNERS.
The following are additional areas for improvement: 1) Presenting the data more clearly, including means and standard deviations. It would also be useful to include RER data, or alternatively to calculate energy cost of running (see Fletcher, Esau, & MacIntosh, 2009. Economy of running: beyond the measurement of oxygen uptake). 2) Clarifying and providing more details on some of the methods, including determination of VT, perhaps a figure to show the testing protocol, and more details on the statistical analysis. 3) More robust citation throughout. 4) Correct minor grammatical errors and inconsistencies in units (ml/km/min some places, ml.km-1.min-1 others, for example).
MEANS AND SD ADDED, ALONG WITH RER DATA. FROM OUT STANDPOINT, USING GAS EXCHANGE DATA WITH RER>1.0 PRESENTS A PARTICULAR PROBLEM RELATIVE TO ESTIMATING THE ENERGY COST OF ACTIVITY 14. ACCORDINGLY, WE CHOSE TO CONSTRAIN OUR MEASURMENTS TO VELOCITIES THAT ALLOWED A STEADY STATE VO2 WITH AN RER <1.0.
See below for further line-by-line comments.
Abstract
Line 13 – remove “in” prior to “which”
DONE
Line 14-15 – make clear that variations need to be at a an exercise intensity below VT
DONE
Line 16 – remove or replace “in the range of 10km-marathon pace” as the speeds tested equate to roughly 7:15 min/mile for the men, which is an easy run pace for a collegiate distance runner
DONE
Line 19 – the use of the term “coefficient of variation” led me to believe that the variations in pace would involve a more complex fluctuation scheme. It would be more clear to simply say that paces were set to alternate each minute above and below 90% of VT by 1.4, 4.2, and 7.0%.
WE HAVE CHOSEN TO RETAIN THE CV AS A MEASURE OF VARIABILITY OF PACE SINCE WE USED IT IN PAPERS THAT PROVIDE THE RATIONALE FOR THIS STUDY. 3,4
Include how VT was determined, and why this method was used.
DONE, WE ROUTINELY USE THE COMMON VO2 VALUE DEFINED BY THE V-SLOPE AND VENTILATORY EQUIVALENT MEASURES. IN OUR HANDS, THESE TWO MEASURES TYPICALLY AGREE QUITE WELL.
Introduction
As noted above, the framing of the paper around world record performances is not appropriate given the methodology employed in this study. The introduction could use a paragraph on VT, as the use and determination of VT is a complicated topic in and of itself (VT1 vs. VT2, and the various methods of determining these thresholds). See the following for starters…
Ekkekakis, P., Lind, E., Hall, E. E., & Petruzzello, S. J. (2008). Do regression-based computer algorithms for determining the ventilatory threshold agree? Journal of sports sciences, 26(9), 967-976.
Faude, O., Kindermann, W., & Meyer, T. (2009). Lactate threshold concepts. Sports Medicine, 39(6), 469-490.
WE ROUTINELY USE THE POINT OF AGREEMENT OF THE V-SLOPE AND VENTILATORY EQUIVALENT METHODS AS A MARKER FOR THE FIRST VENTILATORY THRESHOLD. EXERCISE ABOVE THIS POINT IS NOT NORMALLY ASSOCIATED WITH A STEADY STATE OF VO2 AND AN RER <1, WHICH ARE NECESSARY TO ESTIMATE ENERGY COST FROM RESPIRATORY METABOLISM. WE DO NOT USE A COMPUTER ALGORITHM, BUT RATHER USE A MANUAL INSPECTION OF THE VO2 VS VCO2 AND THE VE/VO2 VS VO2 CURVES TO DERIVE AN ESTIMATE OF VT. WE CHOSE 90% AS A CONSERVATIVE ESTIMATE TO MAKE SURE THAT WE WERE ON A PART OF THE CURVE WHERE VO2 WAS IN STEADY STATE. WE RECOGNIZE THAT THIS IS SLOWER THAN COMPETITIVE VELOICITY IN THE 1 MILE-5KM RANGE, BUT ESTIMATES OF THE EFFECT OF PACE VARIATION BASED ON RESPIRATORY MEASUREMENTS ARE LIMITED BY THE ASSUMPTIONS REQUIRED USING RESPIRATORY MEASURMENTS. WE HAVE TRIED TO EXPLAIN THIS IN THE TEXT.
Methods:
Table 1: Are volumes of training correct? 30 km/week (or even 52 km/week for the men) seems very low for collegiate distance runners. If these are correct, it may be noteworthy and provide some explanation for the relatively poor running economy measured in this group.
YOU ARE CORRECT THAT THESE WERE INACCURATELY TRANSCRIBED. IN THE USA, TRAINING VOLUME IS STILL COMMONLY REPORTED IN MILES/WEEK, AND WE TRANSCRIBED THAT AS IF IT WERE KM/WEEK. TABLE 1 HAS BEEN CORRECTED
Lines 68-69: Include a citation for the VO2max testing protocol, and perhaps a justification for using only treadmill speed instead of incline for the protocol.
IN OUR EXPERIENCE, IT IS VERY COMMON (E.G. NOT REQUIRING CITATION) TO USE SPEED ONLY PROTOCOLS WITH COMPETITIVE RUNNERS, WITH THE USE OF A 1% GRADE TO ACCOUND FOR AIR RESISTANCE. WITH “NORMAL” SUBJECTS, WE WOULD, OF COURSE, HAVE USED A PROTOCOL THAT INCLUDES GRADE INCREMENTS, BUT NEVER WITH COMPETITIVE RUNNERS.
This would also be a good place to include how VT was determined, along with a citation for the method of VT determination that was used.
DONE
Line 80-82: Clarify or consider adding a figure to clearly show the design.
WE HAVE CAREFULLY CONSIDERED THIS, AND DON’T FEEL THAT IT WOULD ADD TO THE CLARITY OF THE PAPER
Lines 86: State the specific model of metabolic system.
DONE
Line 92: Citation for Borg scale should be included.
DONE
Lines 97-99: Include info about software or code used for analysis. Also what was considered statistically significant?
DONE
Figure 1: Make x-axis titles consistent, consider adding dots for individual data to show variation. HR missing units in y-axis label. Y-axes could be re-scaled to better show variation between trials. Add symbol denoting significant difference in RPE figure.
WE CHOSE THE SCALE OF THE Y-AXES TO CONVEY THE RANGE OF OBSERVATIONS CONTRIBUTING TO THE MEAN. IN OUR VIEW, IF SOMETHING IS STATISTICALLY NOT-SIGNIFICNAT, THEN FIGURES SHOULD CONVEY THIS LACK OF SIGNIFICANCE. WE ADDED UNITS TO THE HR FIGURE (ALTHOUGH BPM IS SO UNIVERSAL THAT PUTTING UNITS IN IS UNLIKELY TO BE USEFUL). WE ADDED AN * TO INDICATE THAT STATISTICAL SIGNIFICANCE OF RPE, ALTHOUGH WE HAVE RETAINED THE RANGE OF THE Y-AXIS TO CONVEY THE LACK OF REAL MEANING TO THIS STATISTICALLY SIGNIFICANT RESULT
Lines 113-117: move to discussion.
MOVED
Line 113 – remove “comparatively”
DONE
We are also very curious to see these high CR values. Do you have other data from your lab to place this in context with your system, or other studies using the same system?
TEXT AND REFERENCE ADDED TO SUPPORT INTERNAL VALIDITY OF THE VALUES REPORTED
Discussion
Move discussion away from world record/elite performance, as discussed above.
DONE
Lines 120-129: too much redundant information summarizing the methods. 1-2 sentences would be sufficient before getting to the main findings (currently around line 129).
WE WOULD REALLY LIKE TO RETAIN THIS PART OF THE TEXT, AS WE FEEL IT KEEPS THE PAPER FOCUSED
Line 155: VO2 kinetics is mentioned for the first time – this concept should be included in the introduction and expanded upon here.
VO2 KINETICS SHOULD NOT BE VERY IMPORTANT WITHIN THE CONTEXT OF THIS PAPER, AS AT THE VELOCITIES CHOSEN, VO2 KINETICS ARE VERY RAPID IN WELL TRAINING INDIVIDUALS. FURTHER, THE DESIGN OF THE STUDY WAS THAT EACH SEGMENT WAS RUN WITH THE SLOWER RANGE OF THAT SEGMENT COMPLETED FIRST. THIS WOULD ALLOW ANY SMALL EFFECT OF VO2 KINETICS TO BE ACCOUNTED FOR. LASTLY, ALTHOUGH NOT PRESENTED, THE INDIVIDUAL TIME VS VO2 CURVES FOR EACH SUBJECT SHOWED ONLY SMALL VARIATIONS IN VO2 DURING THE LAST 2 MINUTES OF EACH SEGMENT, TYPICAL OF WHAT ONE NORMALLY SEES DURING INTERVAL TRAINING, FURTHER SUGGESTING THAT THE DESIGN OF THE STUDY MAKES VO2 KINETICS FAIRLY UNIMPORTANT.
Lines 170-171: “velocities consistent with steady state conditions” doesn’t make a lot of sense here, given that the whole ideas is that pace is varying. It seems more accurate to say “speeds consistent with moderate intensity exercise” or “speeds below the ventilatory threshold.”
DONE
Conclusions:
Again, this study does not provide any insight into pacing strategies in WR races due to the much lower intensity of exercise in this protocol.
TEXT AT END OF MANUSCRIPT CHANGED TO REFLECT THIS

Reviewer 2 Report
Summary
The purpose of this study was to determine if changes in running pace effected the cost of running in competitive runners. The study had 20 participants. There were two trials: 1) VO2max and VT determination; 2) 4, 6-min running intervals with 2 min of walking rest. The interval speeds varied: 1) controlled speed & other three intervals had the pace set above and below the average speed each minute. The study found that RPE was higher for the pace variation at 7.0% CV. No other differences in physiological measures were found.
Study Importance
By altering the pace around the average pace makes this study more applicable to real-world running as races are not run on treadmills.
General Comments
Ensure you are consistent with your unit use and format
Ensure all equipment used for the study are specifically stated in the in the methods section shortly after they appear. Specific models will help with repeatability of the study
Curious, did gender affect the results?
Recommend framing the introduction and conclusion around collegiate athletes since the study concerned a D3 population. Discussing world record times is a bit of a reach unless these world record times are in reference to collegiate events.
Recommend expanding the discussion to include why some results were not significantly different among the conditions and limit the repeating of results in the discussion’s first paragraph.
Specific Comments
Line 64: Table 1. The value for weight in the All column is misaligned. Unit format are different from the text
Line 73: 90% VT is mentioned but the methods do not describe how VT was measured. Here is a useful reference: Binder 2008 Methodological approach to the first and second lactate threshold in incremental cardiopulmonary exercise testing
Line 92: Need a reference for the Borg scale
Line 96: Data analysis – please provide the statistical software used for the RM ANOVA & post-hoc tests and the alpha value used to set significance
Line 108: Figure 1 – please ensure this figure is all on one page. Though I feel the y-axes could be modified to limit the long bars on the top figures and the large white space in the bottom figures. This could help make the statistical difference observed for RPE for 7.0% CV more obvious. An asterisk for this data point in the figure would also be helpful for the reader. This will require a notes section below the figure to indicate what the asterisk means. Ensure units are used where needed in the individual graphs
Author Response
Reviewer 2 Comments and Suggestions for Authors
Summary
The purpose of this study was to determine if changes in running pace effected the cost of running in competitive runners. The study had 20 participants. There were two trials: 1) VO2max and VT determination; 2) 4, 6-min running intervals with 2 min of walking rest. The interval speeds varied: 1) controlled speed & other three intervals had the pace set above and below the average speed each minute. The study found that RPE was higher for the pace variation at 7.0% CV. No other differences in physiological measures were found.
Study Importance
By altering the pace around the average pace makes this study more applicable to real-world running as races are not run on treadmills.
General Comments
Ensure you are consistent with your unit use and format
DONE
Ensure all equipment used for the study are specifically stated in the in the methods section shortly after they appear. Specific models will help with repeatability of the study
DONE
Curious, did gender affect the results?
NO, WE RAN A PRELIMINARY ANALYSIS WITH GENDER INCLUDED AS A CO-VARIANT. IT WAS NOT IMPORTANT, AND INDEED THERE IS LITTLE REASON TO SUSPECT A GENDER EFFECT RELATED TO THE CR AT DIFFERENT SPEEDS, SO WE ANALYZED THE STUDY WITH ALLL SUBJECTS BEING INCLUDED AS “WELL TRAINED RUNNER”. THIS IS INDICATED IN THE TEXT.
Recommend framing the introduction and conclusion around collegiate athletes since the study concerned a D3 population. Discussing world record times is a bit of a reach unless these world record times are in reference to collegiate events.
WE CHANGED THE INTRODUCTION TO TRY TO FRAME THE PAPER BETTER. THE ORIGINAL IDEA GREW OUT OF OUR PAPER ON CHANGES IN CV IN WR RACES, AND WAS BOLSTERED BY THE “BEATING YOURSELF” PAPER (WHICH INCLUDED BOTH ELITE AND RECREATIONAL RUNNERS). THIS IS COMPLICATED BY THE NEED TO USE A RELATIVELY SLOW (SUB VT) VELOCITY TO TEST THE IDEA ABOUT THE CR CHANGING, EVEN THOUGH THIS VELOCITY IS MUCH BELOW THE VELOCITY OF INTEREST. HOPEFULLY THE NEW TEXT IS CLARIFYING
Recommend expanding the discussion to include why some results were not significantly different among the conditions and limit the repeating of results in the discussion’s first paragraph.
WE HAVE TALKED ABOUT THIS, AN WOULD LIKE TO KEEP IT MUCH AS IT IS. IN FACT, ONLY THE RPE WAS SIGNIFICANT, AND THAT ONLY 3.0 VS 3.2. IT IS A CASE WHERE SOMETHING IS STATISTICALLY SIGNIFICANT BUT PROBABLY MORE OF AN ARTIFACT THAT A REAL FINDING.
Specific Comments
Line 64: Table 1. The value for weight in the All column is misaligned. Unit format are different from the text
IT IS ALLIGNED AROUND THE +/- NOTATION, WE BELIEVE THIS IS CORRECT
UNIT NOTATIONS EDITED
Line 73: 90% VT is mentioned but the methods do not describe how VT was measured. Here is a useful reference: Binder 2008 Methodological approach to the first and second lactate threshold in incremental cardiopulmonary exercise testing
WE HAVE DISCUSSED THE MEASUREMENT OF VT, AND INCLUDED A REFERENCE THAT REFLECTS THE PRACTICE IN OUR LABORATORY
Line 92: Need a reference for the Borg scale
DONE
Line 96: Data analysis – please provide the statistical software used for the RM ANOVA & post-hoc tests and the alpha value used to set significance
DONE
Line 108: Figure 1 – please ensure this figure is all on one page. Though I feel the y-axes could be modified to limit the long bars on the top figures and the large white space in the bottom figures. This could help make the statistical difference observed for RPE for 7.0% CV more obvious. An asterisk for this data point in the figure would also be helpful for the reader. This will require a notes section below the figure to indicate what the asterisk means. Ensure units are used where needed in the individual graphs
DONE
Round 2
Reviewer 1 Report
The manuscript has been substantially improved and clarified.